# Posttranslational Modification of Human Cytochrome CYP4F11 by 4-Hydroxynonenal Impairs ω-Hydroxylation in Malaria Pigment Hemozoin-Fed Monocytes: The Role in Malaria Immunosuppression

**DOI:** 10.3390/ijms241210232

**Published:** 2023-06-16

**Authors:** Oleksii Skorokhod, Vincenzo Triglione, Valentina Barrera, Giovanna Di Nardo, Elena Valente, Daniela Ulliers, Evelin Schwarzer, Gianfranco Gilardi

**Affiliations:** 1Department of Life Sciences and Systems Biology, University of Torino, 10123 Torino, Italy; vincenzo.triglion@edu.unito.it (V.T.); giovanna.dinardo@unito.it (G.D.N.); 2Department of Oncology, University of Torino, 10126 Torino, Italy; v.barrera@liverpool.ac.uk (V.B.); elena.valente@unito.it (E.V.); daniela.ulliers@unito.it (D.U.); evelin.schwarzer@unito.it (E.S.); 3Department of Eye and Vision Science, University of Liverpool, Liverpool L7 8TX, UK

**Keywords:** malaria, human monocytes, lipoperoxidation, 4-hydroxynonenal 4-HNE, hydroxyeicosatetraenoic acids HETEs, cytochrome P450, CYP4F11

## Abstract

Malaria is a frequent parasitic infection becomes life threatening due to the disequilibrated immune responses of the host. Avid phagocytosis of malarial pigment hemozoin (HZ) and HZ-containing Plasmodium parasites incapacitates monocyte functions by bioactive lipoperoxidation products 4-hydroxynonenal (4-HNE) and hydroxyeicosatetraenoic acids (HETEs). CYP4F conjugation with 4-HNE is hypothesised to inhibit ω-hydroxylation of 15-HETE, leading to sustained monocyte dysfunction caused by 15-HETE accumulation. A combined immunochemical and mass-spectrometric approach identified 4-HNE-conjugated CYP4F11 in primary human HZ-laden and 4-HNE-treated monocytes. Six distinct 4-HNE-modified amino acid residues were revealed, of which C260 and H261 are localized in the substrate recognition site of CYP4F11. Functional consequences of enzyme modification were investigated on purified human CYP4F11. Palmitic acid, arachidonic acid, 12-HETE, and 15-HETE bound to unconjugated CYP4F11 with apparent dissociation constants of 52, 98, 38, and 73 µM, respectively, while in vitro conjugation with 4-HNE completely blocked substrate binding and enzymatic activity of CYP4F11. Gas chromatographic product profiles confirmed that unmodified CYP4F11 catalysed the ω-hydroxylation while 4-HNE-conjugated CYP4F11 did not. The 15-HETE dose dependently recapitulated the inhibition of the oxidative burst and dendritic cell differentiation by HZ. The inhibition of CYP4F11 by 4-HNE with consequent accumulation of 15-HETE is supposed to be a crucial step in immune suppression in monocytes and immune imbalance in malaria.

## 1. Introduction

Malaria is still an important and potentially fatal infectious disease. The hallmarks of malaria are excessive inflammation on the one hand, and incapacitation of innate and adaptive immune responses on the other. The *Plasmodium* parasite grows in red blood cells (RBC) of the human host, where it proteolytically degrades more than half of host cell hemoglobin with the deposition of the heme moiety as insoluble nanocrystal hemozoin (HZ) [1]. Parasite growth and hemoglobin degradation are accompanied by lipid peroxidation [2]. Important detected lipoperoxidation final products are 4-hydroxynonenal (4-HNE) and hydroxy-fatty acids (OH-FAs). The latter derived mostly from lipoperoxidation of arachidonic acid (HETEs) and linoleic acid (HODEs). HETEs and HODEs increase significantly in parasitized RBCs (pRBC) in the mature, HZ-containing stages, and such products are found highly concentrated in native HZ [2]. HZ is expelled from host RBCs, with parasite egress coming immediately in contact with immune cells. Monocytes are crucial for innate and adaptive immunity [3,4]. HZ-containing monocytes are regularly seen in malaria patients’ blood [5,6,7]. The quantity of ingested HZ is relatively high: the recognition and avid phagocytosis by monocytes or granulocytes occurs in circulation or in ex vivo experimental models with isolated monocytes, and is estimated as 6–10 RBC equivalents per monocyte [8,9,10]. HZ and pRBC phagocytosis goes along with the ingestion of the lipoperoxidation products. As a result, crucial immune functions, such as further phagocytosis, oxidative burst, antigen presentation, cell motility, maturation to dendritic cells, and cytokine secretion of the monocyte, are impaired by the deposit of HZ after its phagocytosis [4,10,11,12,13]. 4-HNE and 15-HETE are mechanistically involved in the functional inhibition of HZ-laden monocytes [8,10,13,14]. Thus, eliminating the bioactive products by cell metabolism becomes crucial for tuning the immune functions of monocytes. Monocytes not only have to cope with external, parasite-derived OH-FA and 4-HNE, but also with the internal lipoperoxidation. This process is driven firstly by the extraordinary strong oxidative burst with excessive reactive oxygen species (ROS) production during HZ recognition by TLR4 (toll-like receptor-4, which belongs to the pattern recognition receptor (PRR) family and able to induce oxidative storm in the monocytes) and subsequent phagocytosis [8,15]. Secondly, the long-term oxidative product accumulation is prompted by the heme-catalysed peroxidation due to HZ persistence in the lysosomes [8,11]. A long-lasting steady lipoperoxide rise was described that followed the strong initial increase after HZ-phagocytosis in monocytes [8]. Nonetheless, monocytes are first line defenders and usually maintain their functionality under rather harsh pro-oxidant conditions. Their functional deficit caused by HZ could be the consequence of an impaired or limiting deactivation pathway of the abundantly generated HETEs and HODEs. It is therefore interesting to identify metabolic bottlenecks in the protection against cellular OH-FA accumulation.

An important role of cytochromes P450 in these processes is beginning to emerge. For example, it is known that CYP4 enzymes allow the inactivation of HETEs by ω-hydroxylation [16,17], but their involvement in immune processes is mostly studied in arachidonic acid ω-hydroxylation to 20-HETE or leukotriene metabolism [18,19,20]. The involvement of the nuclear receptor RXR and NF-kB in CYP4F11 expression has been described in immortalized keratinocyte and hepatocyte lines [21,22]. In another study, some CYP4 family enzymes have been studied in polymorphonuclear leukocytes (neutrophils), but, specifically, CYP4F11 was not considered by authors [23]. The expression and transcriptional regulation by PPARγ (peroxisome proliferator-activated receptor gamma transcription factor) of another interesting CYP4 enzyme, CYP4V2, have been shown in the human THP1 immune cell line [17]. Very similar to CYP4F11, CYP4F13 was described to be upregulated during the immune response in mice [24], suggesting the involvement of CYP4 in immunity. Nevertheless, to this date, HETEs’ oxidation by CYP4 enzymes in immune cells has not been studied, despite the prominent role of HETEs and PPAR receptors in immunomodulation [20]. In addition, studies of the role of CYP4 in controlling the cellular HETE levels in human models are missing, and the role of CYP4 enzymes in malaria has not been investigated.

Another lipoperoxidation product 4-HNE is known to rise in HZ-fed monocytes [12]. 4-HNE is able to modify protein function and activity by post-translational covalent binding to lysine, histidine, and cysteine residues as Michael adducts or Schiff bases [10,12,25,26]. Some studies identified CYP enzymes as targets for 4-HNE modifications in humans, mice, and rats [27,28]. These studies suggest a role for 4-HNE in eliciting a functional P450 damage, particularly when substrate binding sites or iron-binding cysteines are involved in the modification.

Here, we check the hypothesis regarding whether the modification of CYP4F11 by 4-HNE offers a mechanistic clue for the persistent incapacitation of immune functions in HZ-fed monocytes. The supposed inhibition of CYP by 4-HNE likely results in delayed HETEs degradation, and the accumulation of the bioactive HETE molecules may affect the regular response in immune cells. For this, enzymatically active recombinant CYP4F11 was expressed, and its HETE-oxidizing activity and the inhibition by 4-HNE was characterized. The appearance of 4-HNE conjugated CYP4F11 was explored in human HZ-fed monocytes, and functional loss of these cells was recapitulated in unfed monocytes by 15-HETE supplementation.

The inactivation of CYP4F11 by 4-HNE conjugation under inflammatory conditions and malaria with consequent delayed HETE degradation might offer a mechanistic explanation for the persistent inhibition of immune functions in HZ-fed monocytes and modulation of cellular immunity in malaria.

## 2. Results

### 2.1. 4-HNE-Protein Conjugate Formation in Primary Human Monocytes Fed with HZ

The correct monocyte phenotype and absence of apoptosis were confirmed by FACS. The increase in 4-HNE levels in monocytes fed with HZ was detected by measuring 4-HNE-protein conjugates at 24 h after phagocytosis. 4-HNE-surface protein conjugates were 2.2 ± 0.5 times higher in HZ-fed and 4.0 ± 0.5 times higher in 4-HNE-treated monocytes than in untreated unfed control monocytes, as detected by FACS (Figure 1A). Latex beads-fed monocytes were not positive for 4-HNE-conjugation with values of 0.8 ± 0.3 vs. unfed control monocytes.

Similar results were observed by fluorescence microscopy with bright 4-HNE positive monocytes after HZ-phagocytosis or 4-HNE treatment, whereas unfed control or inert latex beads-fed monocytes did not visibly accumulate 4-HNE and did not become fluorescent (Figure 1B). The 4-HNE-protein conjugates on HZ-fed and 4-HNE-treated monocytes were 3.3 ± 0.2 and 3.7 ± 0.5 times higher, respectively, as compared to untreated controls (Figure 1C). Latex-fed cells show a 1.1 ± 0.3-fold value of control monocytes (Figure 1C).

### 2.2. CYP4F11-4-HNE Conjugates in HZ-Fed Monocytes and Conjugation Sites in CYP4F11

Following 4-HNE formation and conjugation with cell surface proteins by HZ, the functionally relevant proteins modified by 4-HNE in HZ-fed monocytes were investigated. Proteins from lysed monocytes were electrophoretically separated leading to a series of protein bands evenly distributed between 10–150 kDa, with 4-HNE conjugates detected below 70 kDa (Figure 2). Differences in conjugation between HZ-laden and control cells are clearly visible from 50 to 60 kDa (Figure 2A,B). The only exception is actin at 42 kDa, which is a known target for 4-HNE in HZ-fed monocytes [10,29], and unknown band at 20 kDa.

MALDI-TOFF analysis of SDS-PAGE separated proteins in the molecular weight range from 50 to 60 kDa identified, among others, as CYP4F11 (UniProt Q9HBI6) in unfed control, latex-fed, HZ-fed, and 4-HNE treated monocytes. Specific cysteine, histidine, and lysine residues of CYP4F11 were found to be conjugated with 4-HNE forming Michael adducts in HZ-laden but not in untreated control monocytes. These residues are C45 in the fragment VLAWTYTFYDNCRR (34–47aa), C260, H261 from ACHLVHDFTDAVIQER (259–274aa), H347, C354 from HPEYQEQCR (347–355aa), and K451 from FDQENIK (445–451aa). The location of these residues in the model of the folded CYP4F11 (AlphaFold) shows that they are generally present in the exposed regions of the enzyme (Figure 3). In particular, literature reports indicate that C260 and H261 are located in the substrate recognition site of the enzyme [30], suggesting a possible functional alteration upon 4-HNE modification (Figure 3B). The same modifications were detected in 4-HNE-treated monocytes.

### 2.3. In Vitro Production of Functional CYP4F11

In order to test whether the functionality of the CYP4F11 enzyme is affected by 4-HNE conjugation, CYP4F11 was cloned in a specific vector and heterologously expressed in *E. coli.* The yield of the purified enzyme was 6 mg enzyme/L of bacteria culture, and the presence of the enzyme was confirmed by the prominent protein band at 54 kDa observed on SDS-PAGE (Figure 4A). The typical absorbance spectrum of the purified enzyme is shown in Figure 4B. As expected from a pure and active P450, the Soret peak at 418 nm shifts to 450 nm after reduction and CO binding (Figure 4B,C). The subtraction of the absorbance values obtained for reduced CYP4F11 from CO-bound reduced CYP4F11 was conducted and plotted as “difference spectrum” for clear visualisation of 450 nm peak increase and 418 nm peak decrease (Figure 4C).

### 2.4. Enzyme Modification by 4-HNE

Purified CYP4F11was incubated with 4-HNE at 0.1–1 mM concentrations. In order to confirm the conjugation with 4-HNE, SDS-PAGE was performed in duplicate with the treated CYP4F11 preparation to obtain two identical gels, one for total protein visualisation, the other for detection of 4-HNE-protein-conjugate using specific antibodies after protein transfer to a PVDF membrane (Figure 5). Modification of the enzyme by 4-HNE following incubation with a 4-HNE concentration of 1 mM is shown in lane 2 of Figure 5B. Biologically plausible concentration of 100 µM of 4-HNE, as previously observed in HZ-laden monocytes in vivo, also led well visible conjugates with CYP4F11 (lane 4, Figure 5B).

### 2.5. Effect of CYP4F11 4-HNE Modification on Substrate Binding

The enzymatic ability of 4-HNE-conjugated CYP4F11 was studied by UV-VIS spectroscopy. The binding of typical CYP4F11 substrates, namely, PA, AA, 12-HETE and 15-HETE, was investigated. Experiments were carried out with both 4-HNE-free and 4-HNE-conjugated CYP4F11. The typical spectrum of the enzyme with the heme peak at 418 nm is shown in Figure 6A. Titration with all four tested substrates led to the typical low to high spin transition of the heme as shown by the decrease in the 418 nm peak with concomitant increase in the 390 nm peak. Figure 6A shows, as an example, the data obtained from titrations with 15-HETE. The difference between the spectra of CYP4F11 with and without substrate clearly illustrates the peak loss at 418 nm and gain at 390 nm with isosbestic point at 405 nm. The values for the apparent dissociation constant Kd were calculated from the difference of absorbance at 390 and 418 nm plotted against the concentration of each substrate. The fitting of the points to a hyperbolic function led to 15-HETE dissociation constant Kd of 73.3 ± 13.5 µM for unmodified CYP4F11 (Figure 6C). As can be seen from Table 1, Kd values of all four tested substrates are in the 38–100 µM range, in the order of HETE concentrations estimated for HZ-fed monocytes based on known 15-HETE values in native HZ and HZ quantities taken up by monocytes. The 4-HNE binding did not change the spectrum of CYP4F11 but interfered strongly with the binding of 15-HETE (Figure 6B,C) and all other three tested substrates to the enzyme. The lack of substrate binding in the tested concentration range suggests the inhibitory effect of 4-HNE conjugation for CYP4F11 activity.

### 2.6. Inhibition of Enzyme Activity of CYP4F11 by 4-HNE

CYP4F11 activity was tested by measuring the NADPH consumption upon substrate addition. Control experiments were carried out in the presence of CYP4F11, substrates, and NADPH but without reductase (CPR). Incubation with CPR with NADPH and substrate, without CYP4F11, did not initiate NADPH consumption. The CYP4F11 enzyme in the presence of CPR and substrate led to NADPH consumption. After incubation, 6.2 ± 1.4% and 6.1 ± 2.2% of NADPH were oxidized if the substrates PA and AA, respectively, were supplemented, reaching 22.3 ± 2.5% and 33.4 ± 8.1% consumption when 12-HETE and 15-HETE were added (Table 2). 4-HNE-conjugated CYP4F11 was inhibited and did not consume significantly NADPH in presence of any of the four substrates (Table 2) as expected from the substrate binding results.

### 2.7. Inhibition of ω-Hydroxylation Activity of CYP4F11 by 4-HNE Modification

The products of CYP4F11 activity were identified by GC. PA and AA standards showed the typical retention time of 14.85 min and 16.91 min after injection in a HP-5 column (Figure 7). The standard for ω-hydroxylated PA was not commercially available, while ω-hydroxylated AA (20-HETE) standard showed the peak at 19.45 min (Figure 7B, insert). Incubation of CYP4F11 with PA, NADPH, and CPR showed a decrease in the substrate peak at 14.85 min and the appearance of the ω-hydroxylation product peak at 15.92 min (Figure 7A, black line). Incubations with AA, under the same conditions, showed a decrease in the substrate peak at 16.91 min and the appearance of the product (20-HETE) peak at 19.45 min (Figure 7B, black line). Control experiments (the same reactions but in the absence of NADPH) did not lead to the presence of the products (Figure 7A,B, red dashed line). Performing the same reaction under same conditions but using the 4-HNE-conjugated CYP4F11, no product was found (Figure 7A,B, green line).

### 2.8. Functional Impairment of Monocytes by 15-HETE: Inhibition of Oxidative Burst and Surface Antigen Expression in Monocyte Derived DC

The 15-HETE accumulation by delayed ω-hydroxylation in monocytes with lacking or suppressed CYP4F11 activity due to 4-HNE conjugation prompted the study of the effect of 15-HETE on immune functions. We tested two very important monocyte features: the ability to induce oxidative burst and the ability to differentiate in dendritic cells.

Supplementing human adherent monocytes with 15-HETE, resulted in a compromised oxidative burst response of cells to N-formylmethionyl-leucyl-phenylalanine (fMLP), a potent monocyte/macrophage activator (Figure 8). The inhibition by 15-HETE was concentration-dependent in the tested biologically relevant concentration range. The significant inhibition was reached at the earliest measuring time point (15 min after HETE supplementation) with any of the tested concentrations: it was 12.7 ± 1.5% at 0.2 µM and 58.6 ± 6.1% at 10 µM, with a highest tested 15-HETE concentration (Figure 8). Reversal of inhibition was observed after 60 and 120 min for 0.2 and 0.4 µM of 15-HETE, respectively, confirming monocytes’ recovery under low 15-HETE concentrations (Figure 8, insert). At higher concentrations, the fMLP-elicited burst activity was persistently impaired within 2 h (Figure 8). Apoptosis (above 2% of monocytes in all conditions) was excluded as the reason for any functional loss.

The expression of functionally relevant antigens on the cell surface of monocyte-derived dendritic cells (DC) was assessed after 7-day differentiation to DC (Appendix A with experimental design scheme). The following surface markers were monitored on DC: major histocompatibility complex (MHC) class II, which presents the antigens derived from extracellular proteins; CD1a and CD83—important regulatory DC molecules; CD40—immune response amplificator; CD64—Fc gamma receptor 1 (FcγRI, FCRI) that binds monomeric IgG-type antibodies with high affinity; and CD54—intercellular adhesion molecule 1 (ICAM-1) that increase in activated immune cells and is important in cellular communication.

Supplementation of monocytes with 15-HETE inhibited, dose dependently, the DC surface antigen expression compared to corresponding control DCs (Figure 9). Inhibition became significant at 10 µM and above, and the expression of MHC class II, CD1a, CD40, CD83, CD64 and CD54 decreased by 30, 61, 44, 45, 57, and 54%, respectively. Most sensible repression showed the IgG-receptor FcγRI CD64, which was significantly decreased even at 1 µM 15-HETE (−24%). 15-HETE recapitulated the persistent inhibition exerted by HZ on DC differentiating from monocytes at day seven after phagocytosis. The inhibition after phagocytosis was confined to HZ uptake, as inert latex beads did not elicit any inhibition after their phagocytosis (Figure 9). The inhibition of expression regards antigen presenting and accessory molecules, Fc-receptors, and molecules that enable intercellular contact, all of which have crucial functional importance for antigen presentation and interaction with lymphocytes, which is essential for cellular and humoral immunity.

## 3. Discussion

Overwhelming inflammation and cellular immunosuppression may explain the occurrence of severe malaria with a potentially fatal outcome and the incapability of the human host to mount sterile immunity against the malaria parasite. Immunosuppression was described for malaria in humans [31,32,33] and animals [34,35,36] as well as in cellular models [37]. Phagocytosis of naturally lipid-coated HZ and HZ-harbouring malaria parasites was crucial for the modulation of oxidative burst, phagocytosis, antigen presentation, cell motility, differentiation of macrophages and dendritic cells, and cytokine production in monocytes [8,12,13,38]. HZ-derived lipid peroxidation products 4-HNE and HETEs [2,8,13,26] target molecules that are crucially involved in the faulty immune responses, such as protein kinase C (PKC), NAPDH oxidase (NOX), NALP3 inflammasome, actin, and surface receptors, and antigen presenting molecules in HZ-laden human monocytes [10,11,26,31,32]. Why a professional phagocyte, that copes with as much as 10 RBC and approximately 30 fmoles fatty acids during a single phagocytic cycle without functional impairment, shows persistent functional loss after HZ uptake and seems not be able to abrogate the immune-suppressive effect of HETEs [20,39] is not yet well understood.

The present study shows for the first time the conjugation of a CYP enzyme with 4-HNE in human primary monocytes after ex vivo phagocytosis of malaria pigment HZ, the natural meal of monocytes during malaria. Phagocytosis of natural HZ by isolated and cultured ex vivo human monocytes is an approved approach to simulate HZ-monocyte interplay in malaria patients [11,40,41]. In this study, HZ phagocytosis was observed by microscopy (Figure 1B, column (1), bright field) as brownish biocrystals inside monocytes in quantities corresponding to those observed in peripheral blood from patients [6,7]. Similar to former investigations on selected proteins [12,14], HZ-phagocytosis resulted in the accumulation of 4-HNE conjugates with cell membrane and cytosolic proteins, as measured by FACS (Figure 1A), fluorescence microscopy (Figure 1B), and immunochemistry (Figure 2). Adequate phagocytosis controls, i.e., unfed monocytes and inert latex beads-fed monocytes, excluded the phagocytosis process per se as 4-HNE source and the observed 4-HNE-protein conjugate generation was restricted to HZ-fed monocytes and monocytes treated with 4-HNE at plausible concentration for malaria pathology. 4-HNE was described to react with lysine, cysteine, and histidine residues of proteins, creating 4-HNE-protein conjugates and provoking functional changes if functionally relevant sites of the protein are affected [25]. Mass spectrometric analysis of 4-HNE-protein conjugates from HZ-fed and 4-HNE-treated monocytes identified CYP4F11 as 4-HNE modified protein (Figure 3). Six amino acid residues were conjugated with 4-HNE (Figure 3), of which the immediately interesting ones were C260 and H261 located in the substrate recognition site of CYP4F11 [30]. Modifications of H347 and C354 are interesting as to their location in a protein segment with similarity to mouse CYP1A2 (Appendix A), that harbours 4-HNE-conjugated H342 [28] and confirms the exposure of this protein segment to 4-HNE attack. A role of the two other modification sites needs to be revealed yet, but the secondary and tertiary structure of the protein are likely changed by the insertion of the nine carbon long chain of 4-HNE on each residue.

The functional relevance of conjugation with 4-HNE was proven with recombinant CYP4F11 is seen in Figure 4. The protein band of the purified enzyme in SDS-PAGE is seen in Figure 4A, and its spectral characteristics resembled the characteristics of CYP4F11 expressed and studied by another group [42]. In vitro conjugation of CYP4F11 by 4-HNE was detectable at physiologically plausible concentrations of 100 µM 4-HNE supplemented to purified CYP4F11 (Figure 5) and blocked the binding of all four tested substrates palmitic and arachidonic acids, 12-HETE, and 15-HETE (Figure 6). To note, the tested substrate concentrations up to 100–160 µM are plausible physiologic concentrations [43] and binding to the unmodified enzyme underpins the role of CYP4F11 in HETE degradation in human monocytes. In vitro activity measurement of CYP4F11 revealed that 12-HETE and 15-HETE were the better substrates for CYP4F11 as compared to palmitic and arachidonic acids, nevertheless the enzymatic activity was strongly inhibited by 4-HNE (97–100%) irrespective which substrate was used (Figure 6, Table 2).

The final evidence for an inhibition of ω-hydroxylation activity of CYP4F11 by 4-HNE was provided by analysing substrates and products of the CYP4F11 catalysed reaction by GC. The palmitic or arachidonic acid as substrates as well as the increase in their ω-hydroxylation decrease in products along the reaction time was completely suppressed when functional CYP4F11 was replaced by the 4-HNE-conjugated enzyme (Figure 7).

Lipid peroxidation is a potent process which modulates numerous pathways and cellular responses and does not just harm by destruction. Lipid peroxidation products 12- and 15-HETEs, produced by the malaria parasite, are dragged by HZ into phagocytes [2,43]. These molecules are of high anti-inflammatory potential, and 15-HETE was shown here to dose dependently inhibit the fMLP-elicited oxidative burst in human monocytes (Figure 8). Inhibition by low 15-HETE concentrations was quickly resolved by the cell. Concentrations above 5 µM, however, which correspond to the HETE concentrations measured previously in phagocytosed trophozoites and expelled HZ [2], irreversibly inhibited monocyte burst activity during the observation period (Figure 8). Those values might be reached in monocytes if the degradation of 15-HETE is impaired and 15-HETE accumulates. Similarly, the long-term presence of 15-HETE at or above 10 µM significantly inhibited the surface antigen expression in monocyte-derived DC and matched with the inhibition due to HZ (Figure 9). Apoptosis was excluded as the reason for any functional loss. Hence, enzymes that metabolize HETEs are of crucial importance for regular immune cell functions, and a decreased enzyme activity will lead to excessive HETE accumulation in the cell with subsequent immunosuppression.

In summary, due to the similarity of 4-HNE modifications found in CYP4F11 from HZ-laden or 4-HNE-treated monocytes and in 4-HNE-treated recombinant CYP4F11 as well as the strong inhibition of the recombinant enzyme by 4-HNE, a loss of CYP4F11 activity in HZ-laden monocytes is plausible. 4-HNE-binding to proteins is usually irreversible and the inhibition should persist in monocytes. The lack of HETE-metabolizing activity results in the excess of HETEs, which was able to impair monocyte oxidative burst and differentiation ability (Figure 8 and Figure 9).

## 4. Materials and Methods

### 4.1. Culturing of Plasmodium Falciparum (Pf) and Isolation of HZ

The study was carried out in accordance with the Declaration of Helsinki and authorised by the Ethical Committee of University of Torino, Italy (identification code 0114178). Informed consent was obtained from all donor subjects involved in the study. Blood from healthy adult donors was obtained from the local blood bank (AVIS, Torino, Italy), treated with heparin, and promptly used for RBC or monocyte isolation. *Pf* parasites (Palo Alto strain, mycoplasma-free) were kept in culture with RBC, and HZ was isolated as described [44].

### 4.2. Opsonisation of HZ and Latex Beads for Phagocytosis

HZ was finely dispersed in 100 µL of phosphate buffered saline (PBS) at 30% (*v*/*v*). Latex beads (0.114 µm diameter) were suspended at 5% (*v*/*v*) in 100 µL of RPMI-1640 medium, and then HZ and latex beads were incubated separately in 100 µL of fresh human AB serum for 30 min at 37 °C.

### 4.3. Isolation of Monocytes, Phagocytosis of HZ and Latex Beads, and Treatment with 4-HNE

Monocytes were isolated from freshly drawn peripheral blood of healthy donors (AVIS, Torino, Italy) by Ficoll centrifugation and lymphocyte depletion [13] with PanT/PanB Dynabeads (ThermoFisher, Waltham, MA, USA) according to manufacturer instructions.

Phagocytosis and cell treatment were started at time 0 by adding either opsonised HZ (50 RBC equivalents, in terms of heme content, per monocyte), latex beads (10 µL of a 100-fold dilution of the opsonised latex bead suspension per million monocytes), or 4-HNE at 10 µM final concentration to adherent monocytes (the scheme of experimental design regarding cellular studies is reported in Appendix A). Note, 4-HNE was activated from (E)-4-hydroxynonenal-dimethylacetal (Merck, Darmstadt, Germany) immediately before experiment to insure the best conditions for 4-HNE treatment, according to the manufacturer’s instructions. Untreated monocytes were kept in parallel as control under the same conditions. After a 3 h incubation, phagocytosis was stopped by three washings with RPMI 1640 medium, when non-phagocytosed material was washed out, and the incubation medium was changed with supplementation of 80 ng/mL granulocyte macrophage colony-stimulating factor (GM-CSF) and 40 ng/mL interleukine-4 (IL-4). The phagocytosis was assessed and confirmed by microscopy in all of the replicate dishes with HZ-fed cells and compared to controls in bright field microscopy at 3 h and 24 h. At least 300 monocytes were analysed and the percentage of phagocytosed monocytes was over 85% (Leica DR IRB fluorescence microscope equipped with a Leica DFC 420C camera, a 63× oil planar apochromatic objective with 1.32 numerical aperture, version 3.3.1 of the Leica DFC image software (Leica Microsystems, Wetzlar, Germany)).

### 4.4. Differentiation of Dendritic Cells (DC) from Human Monocytes

Adherent HZ- or latex fed or unfed control monocytes were kept for 7 days in RPMI 1640 medium supplemented with 80 ng/mL GM-CSF and 40 ng/mL IL-4 as detailed in [13,14]. 15-HETE was supplemented to unfed monocytes at 0–20 µM and was replaced when culture medium was changed each second day.

### 4.5. Flow Cytometry Analysis of Cell Phenotype, Surface 4-HNE-Protein Conjugates and Apoptosis

Cell phenotype was characterized by CD1a, CD14, CD40, CD54, CD64, CD83, and MHC class II detection by flow cytometry (FACS). Conjugates of 4-HNE with cell surface proteins were quantified by anti-4-HNE-conjugate antibody (HNEJ-2, Abcam, Cambridge, UK) at 24 h after treatments by FACS [45]. Eventual apoptosis was tested by FACS (Apoptosis Detection Kit from Merck). MFI values were acquired from 10^5^ cells.

### 4.6. Oxidative Burst Measurement

ROS production of adherent monocytes was measured by luminol enhanced luminescence technique in a Sirius luminometer (Berthold, Pforzheim, Germany) as described [2], with the exception that burst was elicited by supplementing cells with 100 nM (final concentration) of N-formylmethionyl-leucyl-phenylalanine (fMLP) instead of phorbol ester.

### 4.7. 4-HNE-Protein Conjugates Detection by Microscopy

To detect the 4-HNE conjugates on the cell surface by microscopy, adherent monocytes were delicately washed with PBS supplemented with 2% of albumin (PBS-A), incubated with primary anti-4-HNE-conjugate antibody (clone HNEJ-2, Abcam, 1:50 dilution) at room temperature (RT) for 1 h, washed 3 times in PBS-A, and incubated with secondary anti-mouse FITC-conjugated antibody (1:300) at RT for 1 h. After final washing with PBS-A, cells were immediately observed in bright field and in fluorescence green channel with excitation/emission 488/514 nm wavelength from the Ar/Kr laser in a Leica DR IRB fluorescence microscope with Leica DFC 420C camera, using a 63× oil planar apochromatic objective with 1.32 numerical aperture and the version 3.3.1 of the Leica DFC image software.

### 4.8. 4-HNE-Protein Conjugates Detection by SDS-PAGE/Western Blotting (WB)

Lysate proteins from adherent monocytes were subjected at 20 µg/lane to SDS-PAGE/WB. Alternatively, purified CYP4F11 was modified in vitro by 4-HNE (see below) and run alongside with non-modified CYP4F11 in each of two gels: one gel was stained immediately after SDS-PAGE by Pro Blue Safe stain (Giotto Biotech, Sesto Fiorentino, Italy) and the proteins from another gel were transferred to PVDF membrane by WB. The equality of loaded and transferred protein amounts and of protein pattern of all sample lanes was verified by Ponceau S staining. 4-HNE-protein conjugates were detected by anti-4-HNE-conjugate primary antibody (HNEJ-2), HRP-conjugated secondary antibody and ECL assay (Bio-Rad, Hercules, CA, USA). The arbitrary optical density of labelled bands was acquired by ImageLab 4.1 (Bio-Rad) for at least 4 different donors.

### 4.9. Identification of 4-HNE Binding Sites in CYP4F11 by Mass Spectrometry

Identification of CYP4F11 and its 4-HNE binding sites by mass spectrometry was based on method previously reported [10,14]. Adherent monocytes were washed with ice-cold PBS, containing 100 mM glucose, 5 mM mannitol and Complete protease inhibitor cocktail (Roche Diagnostics, Indianapolis, IN, USA), lysed with Laemmli sample buffer, and 20 μg of proteins of different samples were run in a 10% polyacrylamide gel SDS-PAGE. Then, the gels were stained with colloidal Coomassie (18% *v*/*v* ethanol, 15% *w*/*v* ammonium sulphate, 2% *v*/*v* phosphoric acid, 0.2% *w*/*v* Coomassie G-250) for 48 h at RT. Gel slices with the protein bands from Coomassie-stained gels were excised (as thin as possible, including very weak bands), unstained by several washes in 50% *v*/*v* acetonitrile in 5 mM ammonium bicarbonate (AmmB), and successively dried using pure acetonitrile. The gel slices were rehydrated for 45 min at 4 °C in 20 μL of AmmB containing 10 ng/μL of trypsin. Excess protease solution was then removed, and the volume was adjusted using AmmB to cover the gel slices. Trypsin digestion was continued overnight at 37 °C. Samples were loaded onto the MALDI target using 1.5 μL of the tryptic digest mixed 1:1 with a solution of α-cyanohydroxycinnamic acid (10 mg/mL) in 40% *v*/*v* acetonitrile, 60% *v*/*v* of 0.1% trifluoroacetic acid. Mass spectrometry (MS) analysis was performed on a MALDI micro MX spectrometer (Waters, Milford, MA, USA) equipped with a delayed extraction unit, according to the tuning procedures suggested by the manufacturer, operating in reflectron mode.

Peak lists were generated by Protein Lynx Global Server (Waters). The 25 most intense mass peaks were used for database searches against the Swiss-Prot (UniProt) and NCBI databases using the free search program Mascot (http://www.matrixscience.com, accessed on 3 May 2020), introducing 4-HNE as a variable modification from the default Mascot 2.6.00 software list of modifications (Matrix Science, London, UK). Further search parameters included taxa Homo sapiens, trypsin digest, protein molecular mass, monoisotopic peptide masses, two eventual missed cleavages by trypsin, and peptide mass deviation tolerance of 0.5 Da. Identification of protein bands was obtained from triplicate analysis. Considered proteins had a Mascot score higher than 37 for SwissProt and higher than 67 in NCBI searches, suggested by Mascot to be significant.

### 4.10. CYP4F11 Expression and Purification

CYP4F11 was cloned into pCWori (+) vector for the heterologous expression in the *E. coli* DH5α strain. The N-terminal domain of native CYP4F11 was modified by replacing the first 10 amino acids of the wild type sequence (MPQLSLSWLG) by a string of 8 residues (MALLLAVF) [46] and 4xHis-tag was added at C-terminal. The CYP4F11 was purified by ion exchange chromatography coupled with nickel ion affinity chromatography, and eluted with 1 to 40 mM histidine gradient. The pure protein was concentrated by ultrafiltration using centrifugal devices (Amicon, Millipore, Burlington, MA, USA) with a 30 kDa molecular weight cut-off membrane and buffer exchanged to 100 mM potassium phosphate pH 7.4, 20% glycerol, 500 mM NaCl and 1 mM DTT. The enzyme was stored at −20 °C. Before use, the enzyme was thawed, and the storage buffer was exchanged by ultrafiltration to 100 mM potassium phosphate buffer, pH 7.4. Enzyme purity was confirmed by SDS-PAGE.

### 4.11. CO-Binding Spectral Assay

To check the correct folding of the purified cytochrome P450, CO-binding spectral assay was performed at 25 °C. The oxidised CYP4F11 was reduced by sodium dithionite and its spectrum was registered by 89090A UV-VIS spectrophotometer (Agilent, Santa Clara, CA, USA). Then, carbon monoxide was bubbled through the enzyme solution for 30 s and the spectrum of the reduced carbon monoxide-bound form was recorded. The difference spectrum between the CO-bound and the reduced spectra was used to calculate the enzyme concentration by using the absorbance value at 450 nm and molar extinction coefficient ε_450nm_ of 91,000 M^−1^cm^−1^ [47].

### 4.12. Recombinant CYP4F11 Modification by 4-HNE

Incubation of 3 µM CYP4F11 with different amounts (from 0.1 to 1 mM) of 4-HNE in 100 mM phosphate buffer (pH 7.4) was performed at 25 °C for 15 min in 100 µL volume. The excess of 4-HNE was removed by ultrafiltration by Amicon devices and the modified protein was concentrated to be used for 4-HNE conjugates visualisation in WB (see above) and for enzymatic activity tests.

### 4.13. Substrate Binding Assay

The purified CYP4F11 (1.5 µM) was used for titrations with different concentrations of the four substrates palmitic and arachidonic acids (PA and AA), 12-HETE, and 15-HETE in 100 mM phosphate buffer (pH 7.4) at 25 °C. If needed, fatty acids were dissolved in ultrapure ethanol and then diluted in the reaction mixture with a final concentration of ethanol <0.1%. Upon substrate addition, the absorbance (A) spectra were recorded and used to plot the difference of absorbance at 390 nm and 418 nm (ΔA390–418) as a function of substrate concentration. The resulting curve was fitted to hyperbolic equation, and the dissociation constant K_d_ was obtained for each substrate. CYP4F11 modified by 100 µM 4-HNE was also tested for substrate binding under the same conditions as the non-modified enzyme.

### 4.14. NADPH Consumption Assay

NADPH consumption in enzymatic reaction was followed by the decrease in the NADPH absorbance at 340 nm. The reaction mixture contained 10 µM of CYP4F11 enzyme, 10 µM of human cytochrome P450 reductase (CPR), 150 µM of NADPH, and 150 µM of the respective substrate, arachidonic or palmitic acid, 12-HETE or 15-HETE in 0.1 M potassium phosphate buffer, pH 7.4, and the reaction was monitored at 25 °C for 800 s.

### 4.15. Gas Chromatography (GC) Analysis

GC analysis was performed to analyse the correct function of non-modified CYP4F11 enzyme and the inhibition of 4-HNE-conjugated CYP4F11, detecting the products of enzymatic reaction. The enzymatic reaction was performed with 10 µM of CYP4F11, 10 µM CPR, 1 mM MgCl_2_, and 0.1–1 mM of PA or AA in 0.1 M phosphate buffer (pH = 7.4) starting with pre-incubation at 37 °C for 3 min (representative experiments were conducted with 0.1 mM of substrates). Then, 1 mM of NADPH was added, and the reaction was started in the dark at 37 °C for 30 min. The reaction was stopped by adding HCl. The suspension was centrifuged and lipids were extracted form supernatant with methyl tert-butyl ether added at 1:1 (*v*/*v*) under vortexing. After phase separation for 10 min at RT the organic phase from the top was transferred into a new tube, dried with anhydrous MgSO_4_ for 5 min at RT and spun at 2000× *g* for 2 min to remove MgSO_4_ crystals in pellet. The obtained extract was injected in a 7890A GC System (Agilent) with HP-5 column (length 30 m, diameter 0.320 mm, PEG film 0.25 µm, Agilent). The programme was started with an initial temperature of 70 °C and a ramp of 10 °C/min to achieve a final temperature of 300 °C. Analysis was performed by Agilent ChemStation software (Agilent).

### 4.16. Statistical Analysis

The values from at least three independent biological replicates are presented in histograms (means ± SE). Statistical significance was calculated by Mann-Whitney test, indicating *p* < 0.01 by * and *p* < 0.05 by **. At least six different concentrations of the substrates were tested in substrate binding experiments, the average of the five different experiments was considered to fit the data to the hyperbolic function using SigmaPlot 11.0 software to calculate the Kd values.

### 4.17. Data Sharing Statement

The data supporting the findings of this study are available within the article and its Appendix A.

## 5. Conclusions

The conjugation of CYP4F11 enzyme by 4-HNE, raised due to excessive lipid peroxidation in human monocytes in malaria, leads to CYP4F11 inhibition and affects hydroxylated fatty acid metabolism. These findings offer a new mechanism for functional impairment of monocytes and may contribute to immune imbalances observed in malaria.

## Figures and Tables

**Figure 1 ijms-24-10232-f001:**
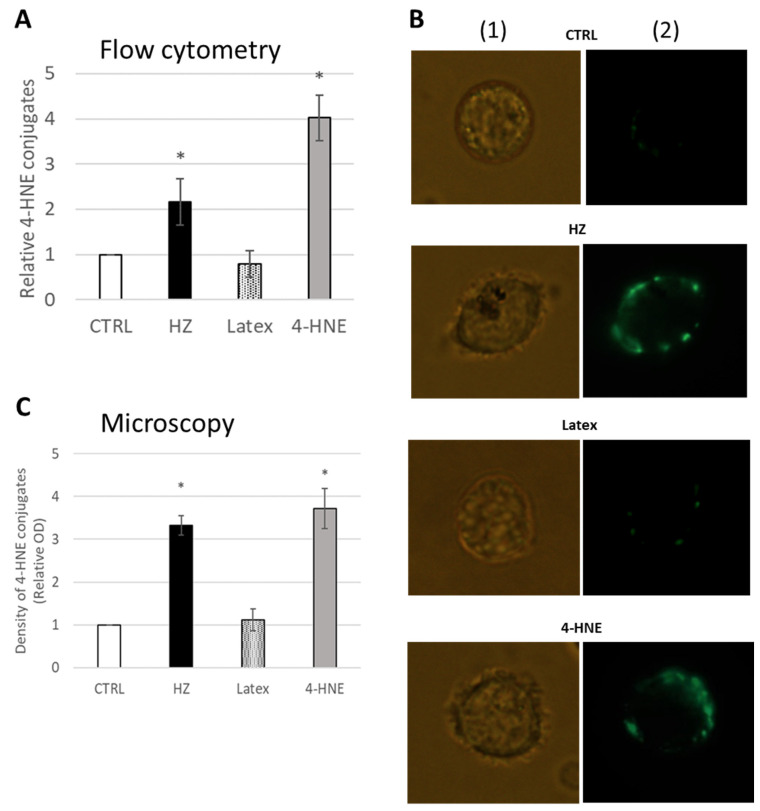
High levels of 4-HNE conjugates on HZ-fed or 4-HNE treated monocyte surface detected by FACS and microscopy. (**A**) Untreated control monocytes (CTRL), HZ-fed (HZ), latex beads-fed (Latex), and 4-HNE-treated (4-HNE) monocytes were assayed for 4-HNE-conjugates by flow cytometry (FACS) with specific anti-4-HNE-conjugate antibodies and FITC-marked secondary anti-mouse antibodies. MFI was measured and referred to MFI of control cells from the same donor. Means ± SE of 3 independent experiments with monocytes separated from freshly drawn blood of three different donors are plotted. The significance of differences between control and treated cells is indicated by * for *p* < 0.01. (**B**) Representative microscopy images of unfed control (CTRL), HZ-fed (HZ), neutral latex beads-fed (Latex), and 10 µM 4-HNE treated monocytes (4-HNE) are shown, acquired in bright field (column (1)) and fluorescence green channel (column (2)) with 488/514 nm excitation/emission wavelengths from a Ar/Kr laser of Leica DR IRB fluorescence microscope with Leica DFC 420C camera, 63× oil planar apochromatic objective with 1.32 numerical aperture, and version 3.3.1 of the Leica DFC image software. (**C**) Statistical analysis was performed after densitometry measurement of green signal intensity collected from at least 50 cells for each of three different donors. Means ± SE are plotted, significance of the difference versus CTRL is indicated by * for *p* < 0.01.

**Figure 2 ijms-24-10232-f002:**
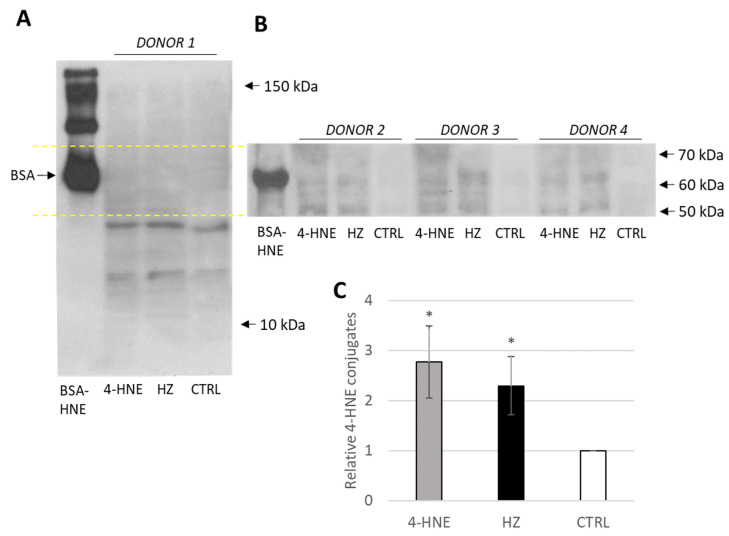
High 4-HNE conjugation with proteins of 50–60 kDa molecular weight in HZ-fed monocytes. Cellular proteins were quantitatively extracted from control untreated (CTRL), 4-HNE-treated (4-HNE) and HZ-fed adherent monocytes (HZ) and 20 µg proteins were separated by Sodium Dodecyl Sulfate Polyacrylamide Gel Electrophoresis (SDS-PAGE) and Western blotted, and anti-HNE-protein conjugates were labelled with a monoclonal primary anti-HNE-conjugate antibody and horseradish peroxidase (HRP)-marked appropriate secondary antibody and visualised by electrochemiluminescence (ECL). The quantity of lysate proteins was determined by the Bradford protein assay (Bio-Rad, Hercules, CA, USA). (**A**) Representative image of 4-HNE-conjugate positive monocyte proteins from donor 1. (**B**) Representative image of 4-HNE-conjugates positive monocyte proteins of 50–60 kDa molecular weight from donors 2–4. Bovine serum albumin, in vitro, conjugated with 4-HNE (BSA-HNE) was included as positive control. (**C**) Statistics of quantified signal from all 4-HNE-conjugate positive proteins, mean ± SE for 4 different donors. * Indicates the significance of difference at *p* < 0.01.

**Figure 3 ijms-24-10232-f003:**
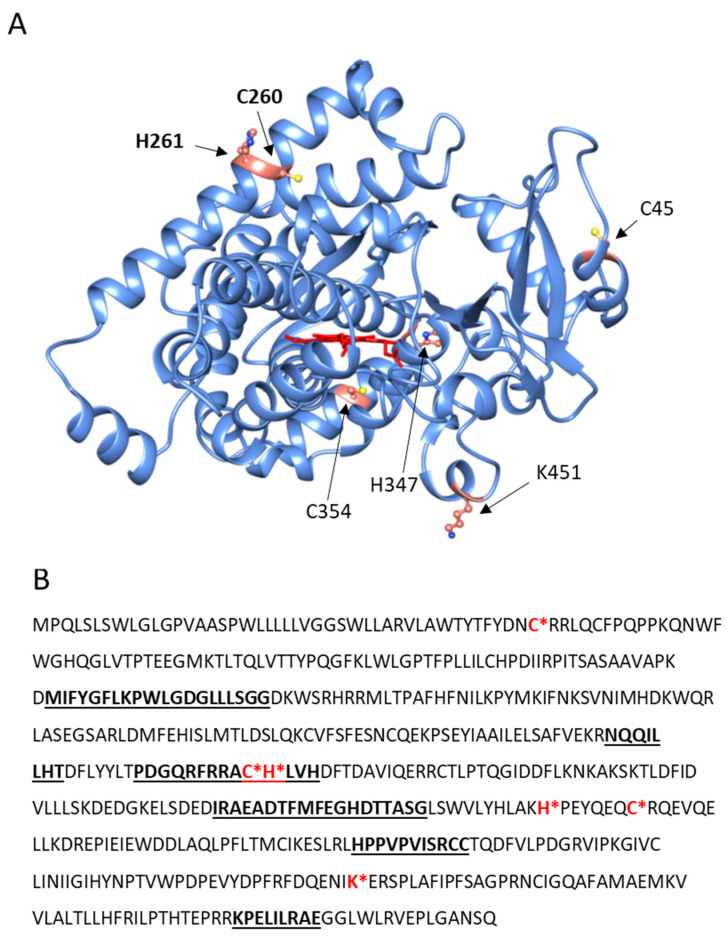
3D AlphaFold representation and primary structure of CYP4F11 with positions of 4-HNE-conjugated amino acid residues and substrate recognition sites. (**A**) 3D representation of the main part of cytochrome CYP4F1 (aa 39–524, the heme is indicated in red) with amino acids modified by 4-HNE indicated by arrows (PDB software www.rcsb.org and AlphaFold model https://alphafold.ebi.ac.uk/entry/Q9HBI6 were used, both accessed 2 May 2023). Two amino acids, C260 and H261, are in bold due to their location in the substrate recognition site. (**B**) Primary structure of CYP4F11 (UniProt Q9HBI6, available online https://www.uniprot.org/uniprot/Q9HBI6, accessed 2 May 2023) with residues modified by 4-HNE (red bold letters and asterisk). The underlined bold strings of residues indicate substrate recognition sites [30].

**Figure 4 ijms-24-10232-f004:**
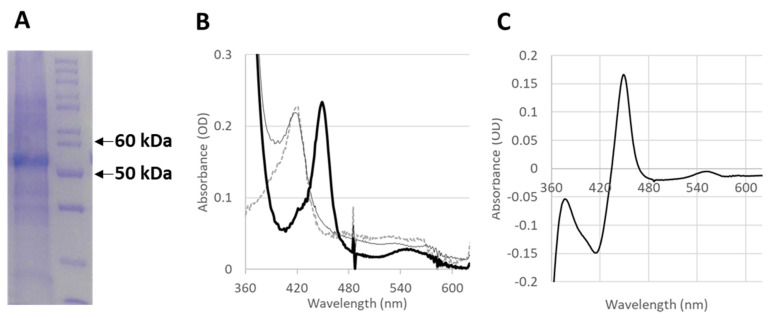
Analysis of spectra and purity of expressed CYP4F11. After CYP expression and purification CO difference spectrum assay was applied. (**A**) Coomassie stained SDS-PAGE gel of purified CYP4F11 (left lane) and molecular weight (MW) markers (right lane). Arrows indicate the 50 and 60 kDa marker protein bands. (**B**) The absorbance spectra of oxidised (dotted line), reduced (thin black line) and reduced CO bound (thick black line) forms of CYP4F11 are shown. (**C**) The difference spectrum of CO-bound reduced vs. reduced CYP4F11 is shown.

**Figure 5 ijms-24-10232-f005:**
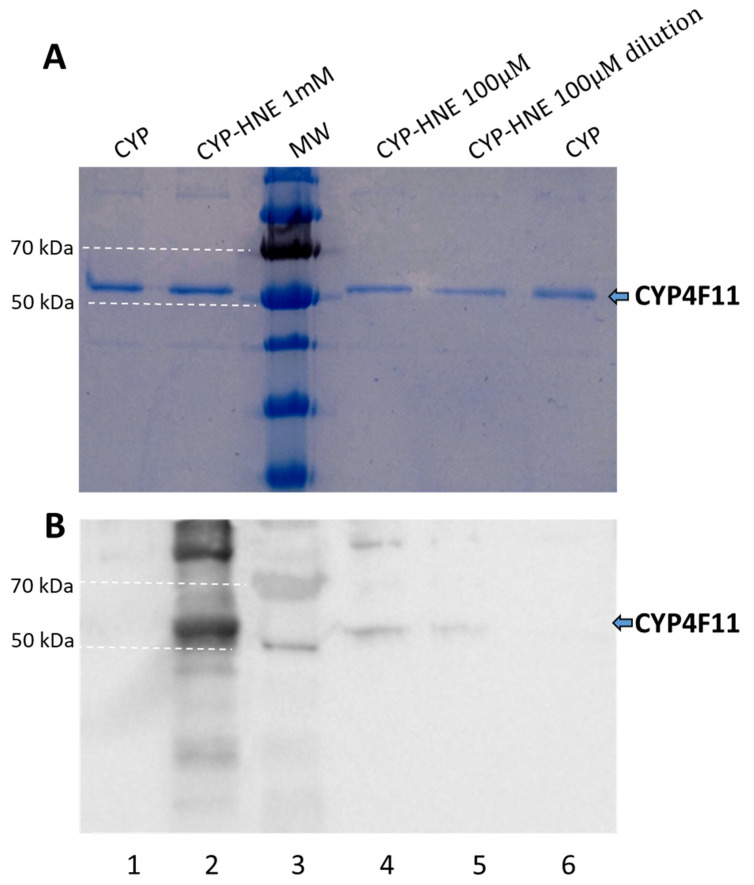
In vitro modification of purified CYP4F11 by 4-HNE. The recombinant CYP4F11 protein preparation was separated in two identical SDS-PAGE gels. (**A**) CYP4F11 protein was visualised in the first SDS-PAGE gel by Pro Blue Safe staining. (**B**) CYP4 F11-4-HNE conjugates were detected by immunostaining of Western blotted proteins from the second gel. For WB, the transferred proteins were probed with specific primary anti-4-HNE conjugate antibodies, appropriate secondary antibodies conjugated with HRP, and evidenced by ECL. Following samples were separated in lane 1: 1.5 µg of CYP4F11; lane 2: 1.5 µg of CYP4F11 treated with 4-HNE (3 µM CYP4F11 + 1 mM 4-HNE); lane 3: molecular weight (MW) markers with dotted white line signed 50 kDa and 70 kDa bands; lane 4: 1 µg of CYP4F11 treated with 4-HNE (3 µM CYP4F11 + 100 µM 4-HNE); lane 5: 0.3 µg of CYP4F11 treated with 4-HNE (3 µM CYP + 100 µM 4-HNE); lane 6: 1 µg of CYP4F11.

**Figure 6 ijms-24-10232-f006:**
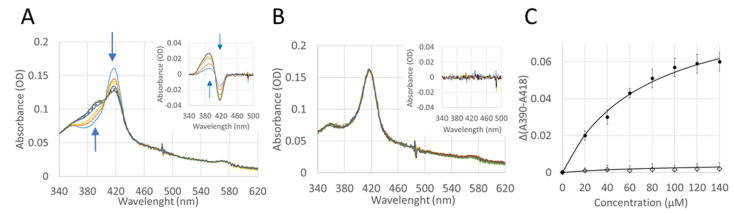
Substrate binding to CYP4F11 is blocked by 4-HNE. (**A**) Typical UV-VIS absorption spectra of CYP4F11 titrated with 15-HETE (15-OH-arachidonic acid) at the concentration 0–140 µM. Difference spectra are shown in the insert. Arrows indicate peak gain or loss at 390 nm and 418 nm, correspondently. (**B**) The spectra of CYP4F11 modified with 4-HNE titrated with the same concentrations of 15-HETE under the same conditions as non-modified enzyme. (**C**) Plot of A390–A418 values observed in the difference spectra versus 15-HETE substrate concentration. Solid circles correspond to values found for the unmodified CYP4F11, open diamonds correspond to data obtained for the 4-HNE modified enzyme. Data are mean ± SD for at least five different measurements. The fitting of the points to a hyperbolic function led to 15-HETE dissociation constant Kd of 73.3 ± 13.5 µM for unmodified CYP4F11.

**Figure 7 ijms-24-10232-f007:**
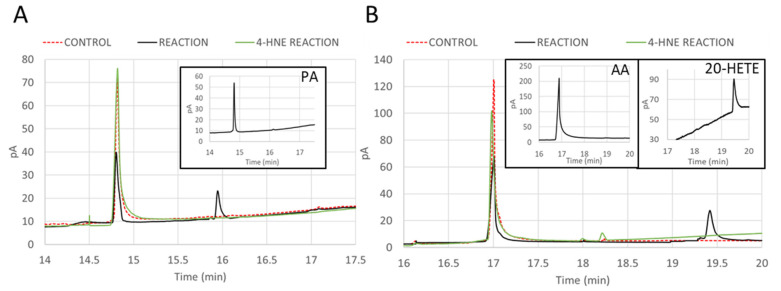
4-HNE inhibits ω-hydroxylation catalysed by CYP4F11. Reaction mixtures of recombinant CYP4F11 with the substrates palmitic acid (PA; panel (**A**)) or arachidonic acid (AA; panel (**B**)) were incubated for 30 min at 37 °C and analysed for ω-hydroxylated fatty acids by gas chromatography (GC). Chromatograms are plotted for tree types of reaction: control reaction where CYP4F11 and substrates were mixed with the missing coenzyme NADPH (CONTROL, red dashed line); complete reaction where mixtures contained all reagents and unmodified CYP4F11 (REACTION, black solid line); experimental reaction under the same conditions as complete reaction but with 4-HNE-conjugated CYP4F11 (4-HNE REACTION, green line). Standards are represented in inserts, with typical peak for PA at 14.85 min (**A**), for AA at 16.9 min (**B**), and for 20-HETE at 19.45 min (**B**).

**Figure 8 ijms-24-10232-f008:**
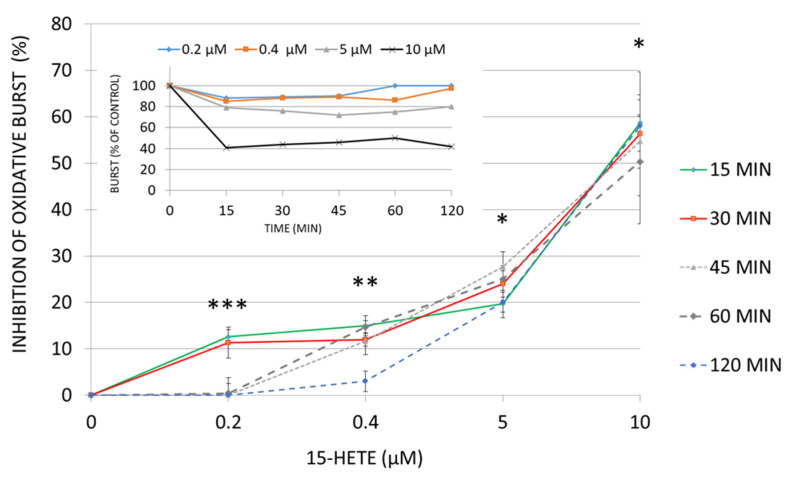
Inhibition of monocyte oxidative burst by 15-HETE. Different concentrations of 15-HETE (1–10 µM) were added to adherent monocytes for 15–120 min, and cellular oxidative burst was elicited by FMLP stimulation. Inhibition of oxidative burst is plotted for different 15-HETE concentrations. Means ± SE are plotted for three different donors, significance of difference vs. untreated control (*p* < 0.01) is indicated by *** for 15 and 30 min exposure; ** for 15–60 min exposure; * for all exposure times. One representative experiment is shown in insert, where the oxidative burst rate is plotted vs. time of 15-HETE exposure.

**Figure 9 ijms-24-10232-f009:**
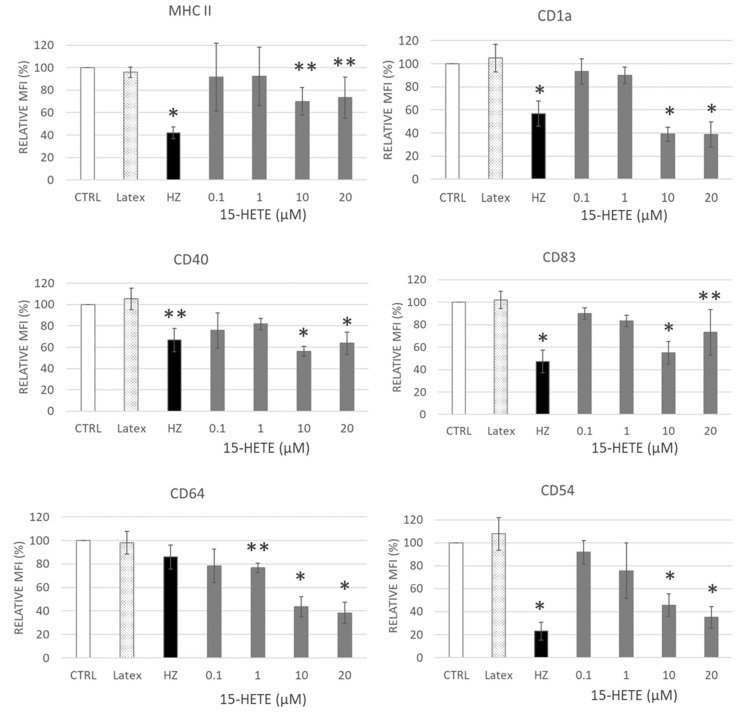
Hemozoin load and 15-HETE treatment impair the expression of functionally relevant antigens on monocyte-derived dendritic cells (DC). Human ex vivo monocytes were treated with neutral latex beads (Latex), malarial pigment hemozoin (HZ), and 15-hydroxyeicosatetraenoic acid (15-HETE, 0.1–20 µM). Surface expression of functionally relevant DC molecules MHC class II, CD1a, CD40, CD83, CD64 and CD54 was measured at day 7 from the start of differentiation by flow cytometry and reported as MFI relative to the expression on untreated cells (CTRL). Mean values ± SE for three donors. Significance of differences for treated cells vs. unfed untreated control (CTRL): *, *p* < 0.01; **, *p* < 0.05.

**Table 1 ijms-24-10232-t001:** Dissociation constants Kd (µM) determined for the different substrates on CYP4F11. Due to the lack of substrate binding on 4-HNE-modified CYP4F11, the Kd were not determined for 4-HNE-modified CYP4F11.

Substrate	Dissociation Constants Kd (µM)
Palmitic acid (PA)	52.1 ± 4.0
Arachidonic acid (AA)	97.8 ± 16.2
12-HETE	38.2 ± 2.5
15-HETE	73.3 ± 13.5

**Table 2 ijms-24-10232-t002:** The rate of NADPH consumption in % by unmodified and 4-HNE modified CYP4F11. NADPH consumption assay was performed for testing the CYP4F11 activity with palmitic acid, arachidonic acid, 12-HETE, and 15-HETE as substrates. NADPH concentrations data were extrapolated from the absorbance at 340 nm measured during CYP4F11 incubation with NADPH, CPR, and substrates at 25 °C for 800 s. Control experiments were performed with CYP4F11, substrates, and NADPH without CPR. Means ± SDs for at least four measurements are presented.

Substrate	Control Experiment	CYP4F11	4-HNE-Conjugated CYP4F11
Palmitic acid (PA)	3.0 ± 0.8	6.2 ± 1.4 *^,§^	3.5 ± 0.7
Arachidonic acid (AA)	2.1 ± 0.7	6.1 ± 2.2 *	3.1 ± 0.9
12-HETE	2.6 ± 1.2	22.3 ± 2.5 *^,§^	3.7 ± 1.3
15-HETE	2.4 ± 1.0	33.4 ± 8.1 *^,§^	4.3 ± 2.1

* Significantly different values in respect to the control experiments at *p* < 0.01. ^§^ Significantly different values in respect to the 4-HNE-conjugated CYP4F11 at *p* < 0.01.

## Data Availability

The data presented in this study are available on request from the corresponding author.

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
