# Peer review of "Posttranslational Modification of Human Cytochrome CYP4F11 by 4-Hydroxynonenal Impairs ω-Hydroxylation in Malaria Pigment Hemozoin-Fed Monocytes: The Role in Malaria Immunosuppression"

_ijms, 2023, doi:10.3390/ijms241210232_

Round 1
Reviewer 1 Report
The manuscript "Posttranslational modification of human cytochrome CYP4F11 by 4-hydroxynonenal impairs ω-hydroxylation in malaria pigment hemozoin-fed monocytes: the role in malaria immuno-suppression" is an important study identifying the role played by the release of hemozoin and immunomodulation by the host. Please find the following comments on the manuscript:
1. Though the authors have tried their best in explaining all the relevant information and background in the introduction, in many places the flow of information is broken and connectivity is lacking. For a reader not very familiar with the subject, it can be difficult to comprehend the language. For example, Line, 54-57, the paragraph is hard to understand, sentence need to be rewritten for clarification.
2. 64-68, no prior information is provided about TLR4, its role, and its connection.
3. 81-85, Incomplete information is provided, what authors want to emphasize by this sentence.
4. Material and Methods: many of the methods are too short, a brief description addition will improve the understanding and relevance. Section 2.15, (Line 258), it will be helpful if authors specify what they plan to analyze with GC and thus tailoring the method will be very useful for readers.
5. In general, the DH5α strain of E.coli is used for cloning, not expression. Can authors verify this?
Author Response
- Though the authors have tried their best in explaining all the relevant information and background in the introduction, in many places the flow of information is broken and connectivity is lacking. For a reader not very familiar with the subject, it can be difficult to comprehend the language. For example, Line, 54-57, the paragraph is hard to understand, sentence need to be rewritten for clarification.
>We apologise for complex explanation, we rewritten now the phrase:
>The quantity of ingested HZ is relatively high: the recognition and avid phagocytosis by monocytes or granulocytes occurs in circulation or in ex vivo experimental models with isolated monocytes, and estimated as 6-10 RBC equivalents per monocyte
- 64-68, no prior information is provided about TLR4, its role, and its connection.
> We added missing information.
>New phrase is:
This process is driven firstly by the extraordinary strong oxidative burst with excessive ROS production during HZ recognition by TLR4 (toll-like receptor-4, which belongs to the pattern recognition receptor (PRR) family and able to induce oxidative storm in the monocytes) and subsequent phagocytosis 8,15 . Secondly, and most likely by the long-term oxidative product accumulation is prompted by the heme-catalysed peroxidation due to HZ persistence in the lysosomes. 8, 11
- 81-85, Incomplete information is provided, what authors want to emphasize by this sentence.
>Here we wanted to collect available information about CYP4 family enzymes.
Now we provided the missing information and more clearly inserted in the contest.
- Material and Methods: many of the methods are too short, a brief description addition will improve the understanding and relevance. Section 2.15, (Line 258), it will be helpful if authors specify what they plan to analyze with GC and thus tailoring the method will be very useful for readers.
>Done. The GC methodological part was improved.
- In general, the DH5α strain of E.coli is used for cloning, not expression. Can authors verify this?
>The E.coli DH5α strain was used for slow CYP4F11 expression, confirming to be efficient in our conditions. Further improvement of the protocol is planned with possible involvement of other E.coli strains.
Reviewer 2 Report
In this study, the authors aimed to investigate the mechanisms underlying the functional impairment of monocytes in severe malaria. They found that excessive lipid peroxidation during malaria led to increased levels of 4-HNE in monocytes. By using the 4-HNE antibody, they identified the target protein CYP4F11, which was conjugated with 4-HNE, resulting in the inhibition of CYP4F11 activity. This inhibition affected the metabolism of 15-HETE and ultimately led to the impairment of monocytes.
The study provides valuable insights into the understanding of how lipid peroxidation and the conjugation of CYP4F11 by 4-HNE contribute to monocyte dysfunction in severe malaria. However, there are a few suggestions to potentially enhance the quality of the manuscript.
1. Could you draw a diagram to describe the mechanism of how 4-HNE and CYP4F11 are involved in the malaria derived monocyte immune imbalances? I think that will impress the readers.
2. Could you insert the full name of ROS, TLR4? It will be helpful to the readers who are not familiar with them.
3. There are some gaps in the result section, could you explain why you perform the experiment following the last result? For example, why did you check the antigen expression? And adding some background information of these antigens may be better. Did you find some antigen expression pattern is different from these in figure 9? Any selective regulation?
4. A space in line 99 is missing.
5. Some typos in line 157, 162, 173, 283.
6. The font size in Figure 3 legend is not uniform.
7. Line 372-373, please add more details to help readers understand what you are doing.
8. In Table 1, 12-HETE should be the ideal substate of CYP4F11 based on the Kd, while 15-HETE was used to evaluate the inhibition of monocyte oxidative burst in figure 8. Could you explain why you chose 15-HETE but not 12-HETE?
The quality of the English language is fine, with only some typos that need to be corrected.
Author Response
- Could you draw a diagram to describe the mechanism of how 4-HNE and CYP4F11 are involved in the malaria derived monocyte immune imbalances? I think that will impress the readers.
Yes. In the graphical Abstract we drawn the diagram representing the 4-HNE and CYP4F11 interaction leading to monocyte long-term suppression.
2. Could you insert the full name of ROS, TLR4? It will be helpful to the readers who are not familiar with them.
Done
- There are some gaps in the result section, could you explain why you perform the experiment following the last result? For example, why did you check the antigen expression? And adding some background information of these antigens may be better. Did you find some antigen expression pattern is different from these in figure 9? Any selective regulation?
As last result we performed the test of monocyte differentiation in dendritic cells, important functional feature of monocytes. To monitor correct differentiation, cell surface expression of numerous markers was monitored. 15-HETE accumulation was shown here as suppressive factor for this process. Expression pattern is shown in Figure 9, confirming firstly the inhibition of differentiation, plausibly due to PPAR-gamma activation by 15-HETE. Possible selective regulation we plan to study in our further research.
We expanded this experimental part in the text, explaining the mean of performed experiment and describing the antigens with background information.
- A space in line 99 is missing.
Corrected
- Some typos in line 157, 162, 173, 283.
Corrected
- The font size in Figure 3 legend is not uniform.
Corrected
- Line 372-373, please add more details to help readers understand what you are doing.
We added the explanation in the line 376.
- In Table 1, 12-HETE should be the ideal substate of CYP4F11 based on the Kd, while 15-HETE was used to evaluate the inhibition of monocyte oxidative burst in figure 8. Could you explain why you chose 15-HETE but not 12-HETE?
We used 15-HETE for experiments shown in Fig 8 for the reason, that 15-HETE is more studied in the literature and it seems more important for immune regulation. Further experiments with 12-HETE are planned.